# Practical Report of Disaster-Related Mental Health Interventions Following the Great East Japan Earthquake during the COVID-19 Pandemic: Potential for Suicide Prevention

**DOI:** 10.3390/ijerph181910424

**Published:** 2021-10-03

**Authors:** Masatsugu Orui, Suzuka Saeki, Shuichiro Harada, Mizuho Hayashi

**Affiliations:** 1Sendai City Mental Health Welfare Center, Sendai 980-0845, Japan; suzuka_saeki_1@city.sendai.jp (S.S.); shuuichirou_harada@city.sendai.jp (S.H.); mizuho_hayashi@city.sendai.jp (M.H.); 2Department of Public Health, Fukushima Medical University School of Medicine, Fukushima 960-1295, Japan

**Keywords:** disaster, mental health, suicide prevention, Great East Japan Earthquake, COVID-19

## Abstract

*Background*: This practical report aims to publicize the ongoing disaster-related mental health interventions following the Great East Japan Earthquake during the COVID-19 pandemic. *Methods*: Disaster-related mental health interventions consisted of: (1) screening high-risk evacuees with high psychological distress (Kessler 6 score ≥ 13) or binge drinking; and (2) visiting selected high-risk individuals and providing them counseling through outreach in evacuee housing. These activity records were compiled from existing material in the Sendai City Office; therefore, no new interviews or questionnaire surveys were conducted. *Results*: During the COVID-19 pandemic, we introduced telephone counseling and shortened the time of support as a result of the restrictions. Counselors addressed issues of “loneliness” or “isolation” among evacuees, who had little connection with society due to the pandemic. Moreover, the procedure for obtaining COVID-19 special financial aid was explained to evacuees in financial difficulty. During this period, the suicide rates in the affected area did not increase significantly as compared to the national average. *Conclusions*: Our report may be instructive in terms of preventing suicide during the pandemic using high-risk approaches and counselors trained in disaster-related mental health interventions.

## 1. Introduction

The Great East Japan Earthquake (GEJE) occurred in March 2011. An unprecedented 9.0 magnitude earthquake generated a massive tsunami that threatened the coastal area facing the Pacific in the Iwate, Miyagi, and Fukushima prefectures in the Tohoku region. The devastating tsunami disaster led to more than 1000 causalities, with the number of evacuees in Sendai City reaching approximately 140,000. The aftermath of the tsunami disaster led to mass evacuations from severely affected areas. These people were forced to live in extremely stressful conditions, facing the loss of their relatives, housing, and/or jobs, while adjusting to the new circumstances [1]. According to previous studies, these devastating natural disasters and their aftermath are known to cause psychological distress in affected individuals [2,3], including post-traumatic stress disorder (PTSD), depression, and suicidality [4,5,6]. Under these conditions, the male suicide rate in the affected area of the Miyagi prefecture exhibited a delayed increase 1.5 years after the occurrence of the disaster [7]. Additionally, several changes in environmental factors (e.g., the termination of free temporary housing) worsened the mental health status of evacuees, even when they were relocated to permanent homes. Consequently, the suicide rates in the affected areas increased once more during the recovery phase [8,9,10,11]. Therefore, mental health support for evacuees was required in order to address the evacuees’ psychological distress and to prevent subsequent suicides. Under the conditions resulting from the GEJE, the Sendai City Suicide Measurement Plan was established in 2019, which focused on supporting the evacuees of the GEJE and taking measures for suicide prevention. The public sector of Sendai City followed this plan and there has since been continuous assistance in the form of numerous disaster-related mental health interventions and suicide prevention measures to support evacuees and affected residents of the GEJE.

In 2020, as a result of the Coronavirus 2019 (COVID-19) outbreak, the Japanese government declared a state of emergency and issued stay-at-home orders; subsequently, the COVID-19 pandemic has affected Japan like every other country. As a result, economic and work activities had to be restricted in all areas without exception, including the aforementioned disaster-related mental health interventions [12]. Therefore, it was expected that evacuees who experienced a devastating disaster would be further psychologically burdened by the social and economic impact of the COVID-19 pandemic [13]. Indeed, in the future, suicide rates in the affected areas may exhibit a further increase. Thus, we continued with the mental health interventions during the COVID-19 pandemic, while applying infection prevention measures and acquiring skills to cope with evacuees with concerns or anxiety regarding COVID-19. The interventions during the COVID-19 pandemic were combined with previous interventions according to the conceptual model (Figure 1).

Our mental health and welfare center is located in Sendai City and was established to promote mental health and welfare among citizens and persons with mental illness, as stipulated by the Act on Mental Health and Welfare for the Mentally Disabled. Similar centers exist in all 47 prefectures and 20 large cities designated by the national government. Moreover, our center is responsible for the improvement of mental health among all citizens, including evacuees of the GEJE and residents affected by the COVID-19 pandemic. Therefore, just after the occurrence of the GEJE, our center established the disaster-related mental health care team, which started providing mental health support to the evacuees of the GEJE and went on to address evacuees’ psychological distress resulting from the COVID-19 pandemic.

This practical report aims to: (1) publicize the status of ongoing disaster-related mental health interventions and human resource development following the GEJE during the COVID-19 pandemic, since practical reports concerning disaster-related mental health interventions during the COVID-19 pandemic are scarce [14]; (2) analyze the suicide rate trend in the area severely affected by the tsunami disaster during the COVID-19 pandemic, while considering the association with disaster-related mental health interventions.

## 2. Methods

### 2.1. Explanation of Subject Area

Sendai City has a population of 1.06 million and is the largest city in the three prefectures affected by the GEJE, namely Miyagi, Iwate, and Fukushima. The areas in which the disaster-related mental health interventions took place were Miyagino Ward (Population: 190,215, as of January 2020) and Wakabayashi Ward (136,465) in Sendai City, which face the Pacific Ocean, and Aoba Ward (292,998), Taihaku Ward (231,353), and Izumi Ward (213,029), which are located in the in-land area of Sendai City. The areas that were heavily damaged by the tsunami were a mixture of agricultural and residential locations. However, other regions had urban functions, such as branch offices of national-level administrative facilities, higher education facilities, and large-scale commercial facilities or factories.

The vast majority of tsunami-disaster evacuees were residents of Miyagino Ward and Wakabayashi Ward; therefore, these were the focus of our disaster-related mental health interventions. Incidentally, some interventions also took place in the in-land area (Aoba, Taihaku, and Izumi Ward), as temporary housing and the restoration of public housing was necessary in these locations (Figure 2).

### 2.2. Practical Activities of Disaster-Related Mental Health Interventions including Human Resources Development and Monitoring Suicide Rates

Disaster-related mental health interventions, including human resources development and monitoring suicide rates, were implemented according to the conceptual model (Figure 3).

### 2.3. Subjects of Disaster-Related Mental Health Interventions

The subjects of disaster-related mental health interventions were evacuees who had been evacuated to temporary housing or public restoration housing in Sendai City, particularly in Miyagino Ward and Wakabayashi Ward.

The procedures required to begin disaster-related mental health support for evacuees are as follows: (1) annual consensual screening of high-risk evacuees in temporary housing or public restoration housing with high psychological distress (Kessler 6 score ≥ 13), binge drinking (e.g., drinking in the morning or daytime on a daily basis), and/or cessation of attending hospitals [15]; and (2) visiting selected high-risk individuals and subsequently confirming their actual status by consulting counselors with health or welfare professional licenses, such as medical nurses, public health nurses, and certificated social workers. After confirmation of their status, if support is needed, continuous counseling is provided through outreach in evacuees’ temporary housing and public restoration housing [15,16].

Incidentally, the percentage of people aged 65 and over in public restoration housing is close to 60%, as societal ageing continues.

### 2.4. Investigation of the of Disaster-Related Mental Health Interventions during the COVID-19 Pandemic

In the case of the GEJE, disaster-related mental health interventions were carried out from an early stage. These disaster-related mental health interventions aimed to support the evacuees in preventing or minimizing the disaster’s psychological distress and help them recover their psychosocial well-being, while also providing support based on outreach, including visits to shelters, temporary housing, and public restoration housing. Moreover, the disaster-related mental health interventions were not limited to individual support: they also sought to contribute to social ties through interaction interventions between the evacuees, including conversation groups and hobby and exercise classes. Because of the number of evacuees in Sendai City as a result of the GEJE, outside volunteers supporting our interventions were limited in the early phases, i.e., up until December 2011 [15,17,18,19].

After completing outside volunteers’ support, our disaster-related mental health interventions supported evacuees while collaborating with counselors working in public health centers in Sendai City (see details in reference [15]).

However, the COVID-19 pandemic has affected Japan just like every other country. Following the outbreak of the pandemic, the Japanese government declared a state of emergency and issued stay-at-home orders in April 2020. Economic and work activities had to be minimized nationwide [12], with measures being implemented without exception. As a result, disaster-related mental health interventions were also forced to temporarily suspend face-to-face consultations at evacuees’ homes. Instead, non-face-to-face support was provided, such as telephone-based consultations, in keeping with the recommendation of the Japanese government. Following the lifting of the state of emergency, disaster-related mental health interventions, particularly those involving visits to evacuees’ homes and face-to-face counseling, were restarted gradually.

Under these circumstances, we investigated the complex situation of disaster-related mental health interventions during the COVID-19 pandemic and compared them with previous interventions.

### 2.5. Comparison of Human Resources Development Regarding Disaster-Related Mental Health Interventions Following the GEJE and during the COVID-19 Pandemic

Human resources development for counselors is essential in order to continue with interventions and maintain their quality; therefore, we also conducted regular training seminars regarding disaster-related mental health interventions. These training seminars for counselors were held after the GEJE to improve counselors’ skills (basic skills concerning coping with traumatic stress and psychological distress, with a particular focus on depression, alcohol-related problems, and suicidal ideation) using case studies and group discussion.

Although we continued conducting human resources development with several restrictions during the COVID-19 pandemic, the style and contents of the training seminars had to change markedly as compared to the previous format. Therefore, a comparison of the pre-COVID-19 scenario and the scenario during the COVID-19 pandemic was conducted.

### 2.6. Trend in Suicide Rates in the Area Severely Affected by the Tsunami Disaster in Sendai City during the COVID-19 Pandemic

To evaluate the mental health status of evacuees in the affected area (Miyagino Ward and Wakabayashi Ward), the suicide rates were monitored following the GEJE, from the acute phase to the recovery phase. In particular, during the COVID-19 pandemic, we evaluated the suicide rates in the area severely affected by the tsunami disaster using a standardized mortality ratio (SMR) and compared them with the rates in the non-affected areas (Aoba, Taihaku, and Izumi Ward) and the national average. The data regarding monthly suicide cases were obtained from the open access website of the Ministry of Health, Labor, and Welfare (2021). The population used as the denominator of suicide rates was the Basic Resident Register Population (provided by the Ministry of Internal Affairs and Communications, Tokyo, Japan).

### 2.7. Ethical Consideration

Ecological studies and health services conducted by the public sector are exempt from ethical review according to the Ministry of Education, Culture, Sports, Science, and Technology and the Ministry of Health, Labor, and Welfare’s epidemiological research guidelines. As a result, an ethics committee review was not required. Incidentally, disaster-related mental health interventions were implemented as part of Sendai City’s health promotion project. These interventions were compiled from existing material in the Sendai City Office. No new interviews or questionnaire surveys were conducted with the residents.

## 3. Result

### 3.1. Disaster-Related Mental Health Interventions during the COVID-19

During the COVID-19 pandemic, our disaster-related mental health interventions were forced to temporarily suspend face-to-face consultations in evacuees’ homes. Instead, non-face-to-face support was provided, such as telephone-based consultation, in keeping with the recommendations of the Japanese government. However, these were also affected by the infection prevention measures (e.g., the wearing of mask, the shortening of the counseling time to less than 10-15 min, and counseling at the front door). In cases of evacuees experiencing economic hardships, we explained the procedure for special COVID-19 cash payments (from the national government) and temporary loan emergency funds (from the Council of Social Welfare). Many elderly evacuees had been living in public restoration housing, and some people had significantly reduced opportunities to go out due to the COVID-19 pandemic. Therefore, leaflets containing recommendations for indoor exercises were distributed to every household in public restoration housing. This distribution also led to an opportunity for interaction between counselors and evacuees.

According to reports from the counselors, the elderly evacuees in the public restoration housing made comments such as “I am anxious for infection at the beginning of the COVID-19 pandemic, so I had to take medication,” “I am refraining from going out because I am worried about how other residents in the restoration housing will see it,” and “I no longer have a relationship like visiting home and talking to my neighbors.” As a result, there was a situation whereby the “loneliness” or “isolation” among evacuees with little connection with society became apparent as a problem. On the other hand, counselors involved in mental health interventions could support evacuees using telephone counseling for shortened periods of time due to infection prevention procedures. A previous study noted social isolation among evacuees who moved to public restoration housing even before the COVID-19 pandemic [20]. Therefore, our interventions could respond promptly to “loneliness” and “isolation” among evacuees. Additionally, the procedure for special cash payments due to the COVID-19 pandemic (from the national government) and temporary loan emergency funds (from the Council of Social Welfare) helped evacuees who faced financial hardship due to restrictions on economic activity.

Finally, the outcomes of the disaster-related mental health interventions during the COVID-19 pandemic were as follows: according to the Sendai City office reports, the number of support sessions per household was unchanged, even during the COVID-19 restrictions (April 2019 to March 2020: 11.6 times per household, April 2020 to March 2021: 10.8 times per household). Incidentally, the total number of visits and telephone counseling sessions for evacuees from April 2020 to March 2021 decreased as compared to the previous period (4986 times as compared to 3841 times). However, there were few changes in the frequency of support sessions to evacuees because the number of households being supported also decreased as compared to the previous period (430 households as compared to 355 households).

### 3.2. Human Resources Development Regarding Disaster-Related Mental Health Interventions during the COVID-19 Pandemic

More than 30 counselors that worked at the public health center in Sendai City attended the training seminars conducted just after the GEJE. All counselors were medical nurses, public health nurses, psychologists, or certified social workers. These training seminars consisted of: (1) an induction seminar regarding disaster-related mental health interventions; and (2) five to six seminars using case studies and group discussions focused on evacuee-specific mental health issues and community approaches. These seminars were held annually after the GEJE. The contents of the counselor training seminar program are shown in Table 1. This style of annual training seminar was held until 2019.

As regards the human resources development during the COVID-19 pandemic, there was not only a need for basic skills to cope with mental health issues, but also additional knowledge of COVID-19 and aspects related to mental health issues caused by COVID-19 (e.g., coping with the anxiety of infection, risk communication with evacuees, and self-care (associated with COVID-19 in particular) while using appropriate media [21,22]). Moreover, it was challenging to hold group discussions as a result of infection prevention measures. Therefore, the counselor training seminar in 2020 and 2021 endeavored to implement online lectures while avoiding face-to-face contact. Furthermore, we tried to continue presenting opportunities to improve counselors’ skills in telephone counseling, which was required during the COVID-19 restrictions. Additionally, alcohol-related problems [23] and social withdrawal [24] among middle-aged evacuees were among the problems in the recovery phase of the GEJE; these topics were included in the online training seminars, which occurred four times (Table 2).

### 3.3. Result of Suicide Rate Trends in the Area Severely Affected by the Tsunami Disaster in Sendai City during the COVID-19 Pandemic

To evaluate the mental health status of evacuees in the area affected by the devastating tsunami disaster, it was considered essential to track the trend of suicide both in the long-term and the recovery phase after the incident. Although it was reported that the suicide rate in the affected areas following the GEJE increased 1–2 years after the disaster [7], there was also a second increase during the recovery phase, which coincided with the termination of the provision of free temporary housing [8,9,10,11]. Therefore, the suicide rate trend in the affected area is of great significance regarding mental health measures in the affected area.

As a result, excepting the year when the GEJE occurred, i.e., 2011, the suicide rate in the affected area continued to decline, equivalent to that of the national average. Moreover, the suicide rates almost synchronized both in the affected and non-affected areas, except in 2011. However, the rate in the affected area did not exceed the national average, even during the COVID-19 pandemic, whereas the suicide rates in the non-affected area rose significantly (Standardized Mortality Rate, SMR: 1.20, 95% Interval Confidence: 1.02–1.47) during the COVID-19 pandemic in 2020 (Figure 4). Incidentally, only the non-affected area in 2009 demonstrated a significantly higher SMR as compared with the national average (SMR: 1.23, 95% Interval Confidence: 1.07–1.41 in the non-affected area in 2009).

## 4. Discussion

This practical report demonstrated that disaster-related mental health interventions continued despite the COVID-19 pandemic; these took the form of telephone counseling and shortened counseling sessions as a result of the adverse conditions created by the COVID-19 restrictions. The suicide rates in the area severely affected by the tsunami disaster did not exceed the national average even during the COVID-19 pandemic, while the rates in the non-affected area rose significantly. In this context, the differences in disaster-related mental health interventions between the pre-COVID-19 scenario and the scenario during the COVID-19 pandemic are shared in this section. Moreover, the suppression of the increase in suicide rates in the affected areas is discussed.

### 4.1. Disaster-Related Mental Health Interventions and Human Resources Development during the COVID-19 Pandemic

Although there were several restrictions regarding the counseling situation (e.g., the wearing of a mask, a reduced counseling time of less than 10–15 min, and counseling at the front door), disaster-related mental health interventions were able to continue and evacuees received support with psychological distress or psychosocial issues even during the COVID-19 pandemic.

In fact, there was positive feedback from one counselor: “I thought that I was able to help evacuees with any concerns or anxiety about COVID-19 and address the issue promptly because there was a disaster-related intervention system already in place.” Moreover, counselors were able to address evacuees who faced financial hardship due to economic activity restrictions while providing them with financial aid procedures. This prompt financial and social support for evacuees was similar to what took place in the period of relocation to public restoration housing [15]; the previous experiences clearly helped counselors react smoothly and promptly to evacuees’ in need.

During the previous human resources development, we mainly trained counselors using case studies. The seminars consisted of deep and careful discussions concerning individual evacuees with mental health issues, including traumatic stress from severe disaster experiences and grief, and community issues such as social isolation and loneliness. However, we had to change to an online seminar style during the COVID-19 pandemic to prevent the spread of the disease. Moreover, we also had to change the contents of the seminars in order to cope with the anxiety related to COVID-19, risk communication, and basic telephone counseling skills. As mentioned above, the differences between the pre-COVID-19 scenario and the scenario during the COVID-19 pandemic were related to dealing with the anxiety caused by the pandemic. In addition, telephone counseling and shortened counseling sessions were introduced to prevent infection. This required additional skills with which to evaluate evacuees’ mental health status by telephone. There are, however, several advantages to telephone counseling, such as being able to provide sufficient time for counseling even under pandemic conditions [14,25,26]. For this reason, we held a specific seminar regarding telephone counseling during the COVID-19 pandemic.

As a result of these efforts, the number of support sessions per household and disaster-related mental health interventions continued without a remarkable decrease, although disaster-related mental health interventions were temporarily unable to take place face-to-face in the evacuees’ homes.

### 4.2. The Trend of Suicide Rates in the Area Severely Affected by the Tsunami Disaster during the COVID-19 Pandemic

Our findings showed that the suicide rate in the affected area did not increase as much as it did in the non-affected area, while national rates increased in 2020 [27]. There was uncertainty regarding the direct reasons for this. One of the factors may be the ongoing disaster mental health interventions that supported evacuees who experienced the GEJE [9], even during the COVID-19 restrictions.

In previous studies, social issues such as hopelessness to live can be risk factors among the elderly [28,29]. Additionally, hopelessness and lack of connectedness to others are two factors that have been associated with increased risk of suicidal thoughts and behaviors [30]. Furthermore, one of the risk factors of suicidality that should be considered during the COVID-19 pandemic is demoralization [31], which results from the unusual experience and can be seen as similar to exposure to very high levels of stress [32]. Therefore, elderly evacuees who experience a disaster can have feelings of hopelessness and reduced connectedness to others, and consequently, this may cause an increase in suicide rates in the affected area [33].

Our interventions focused on responding promptly to cases of “loneliness” and “isolation,” which can potentially lead to “demoralization” in elderly evacuees who likely have feelings of hopelessness and reduced connectedness during the COVID-19 pandemic. Moreover, we were able to provide information regarding financial aids to evacuees who faced financial hardship. Furthermore, our interventions may have helped prevent suicidal behavior associated with problematic drinking, by applying a high-risk approach to binge drinking [34].

As a result, the suicide rates in the affected area did not remarkably increase as compared to those in the non-affected area, because of the support provided for the many evacuees. Under these situations, therefore, ongoing disaster-related mental health interventions based on consensual screening and the high-risk approach (e.g., focused on psychological distress and binge drinking among evacuees) and outreach-based support may have prevented a remarkable increase in suicide rates in the affected area, even during the COVID-19 pandemic: a scenario that generated feelings of loneliness, subsequent depressive states, and binge drinking behavior among the elderly evacuees in the public restoration housing in particular [23,29,35].

### 4.3. The Limitation and Strength of the Present Practical Report

The present study has several limitations. First, it was impossible to grasp the exact reason for the suicide rates in this study; therefore, we could not directly determine the effect of the ongoing disaster-related mental health interventions. Second, other unknown factors, such as economic conditions, may influence the suicide trend in the affected area during the COVI-19 pandemic. As a result of suicide rate analysis, only the non-affected area in 2009 had a significantly higher SMR than the national average, which might be related to the deterioration of the economic condition due to the global economic crisis in 2008 [36]. Therefore, the increased suicide rates in the inland area in 2020 may be associated with the worsening economic conditions caused by the restriction of economic interventions. Unfortunately, it was impossible to fully assess the economic condition of each ward in Sendai City. Therefore, further studies are required to evaluate the association between suicide rates in the affected area and these conditions. Thirdly, the number of subjects of disaster-related mental health interventions in the area severely affected by the tsunami disaster was small as compared to the entire population of the Miyagino and Wakabayashi Wards. Finally, the region of study was restricted to the coastal area of Sendai City in Miyagi prefecture, even though the tsunami disaster following the GEJE caused damage in the Iwate and Fukushima prefectures. However, according to the annual suicide report from the Ministry of Health, Labor, and Welfare, Japan, the number of suicides related to the GEJE, based on suicide notes and family testimonies, was the lowest in 2020, and this included Miyagi, Iwate, and Fukushima prefectures (Ministry of Health, Labor, and Welfare, 2021). Therefore, further studies are required in order to elucidate the association between the suicide rates in the affected area during the COVID-19 pandemic and the disaster-related mental health interventions, including other affected areas.

On the other hand, there were several strengths to this practical report. In fact, the practical reports regarding disaster-related mental health interventions in the mid- to long-term period after the GEJE were limited [8,15,26,37], and the reports during the COVID-19 pandemic even more so [14]. While referring to developing human resources and addressing mental health issues in other GEJE disaster areas [25,37], our interventions could have provided continuous support to evacuees, even during the COVID-19 pandemic restriction period. This is considered a strength of the current practical report, which was able to promptly address mental health and other psychosocial issues among evacuees, even during the COVID-19 pandemic.

## 5. Conclusions

Even during the COVID-19 pandemic, disaster-related mental health interventions, including human resources development, were able to continue, supporting evacuees with psychological distress and/or psychosocial issues, while addressing concerns related to the COVID-19 pandemic. The number of support sessions per household was maintained without a remarkable decrease, despite a temporarily suspension of face-to-face consultations as a result of the COVID-19 restrictions. These interventions may have influenced the suicide rate trend in the area severely affected by the tsunami disaster during the COVID-19 pandemic, which did not exceed the national average, even in the COVID-19 pandemic context, despite the rates in the non-affected area rising significantly. Therefore, our report may provide important lessons for suicide prevention during the COVID-19 pandemic related to the value of implementing high-risk approaches and providing counselors to support evacuees [38] and disaster-related mental health interventions.

## Figures and Tables

**Figure 1 ijerph-18-10424-f001:**
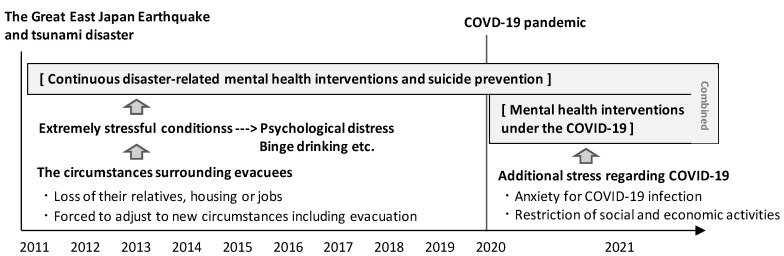
The conceptual model of disaster-related mental health interventions.

**Figure 2 ijerph-18-10424-f002:**
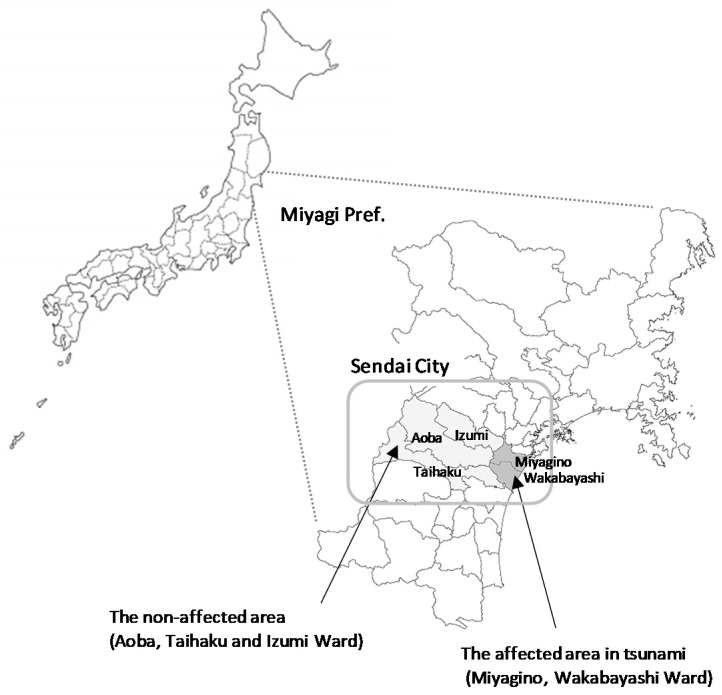
The GEJE and tsunami disaster-stricken areas in Sendai City. In this study, the tsunami disaster-affected area is defined as comprising Miyagino Ward and Wakabayashi Ward, both of which are proximal to the Pacific Ocean in Miyagi prefecture, and appear in dark gray. The non-affected area is defined as Aoba Ward, Taihaku Ward, and Izumi Ward, which appear in light gray.

**Figure 3 ijerph-18-10424-f003:**
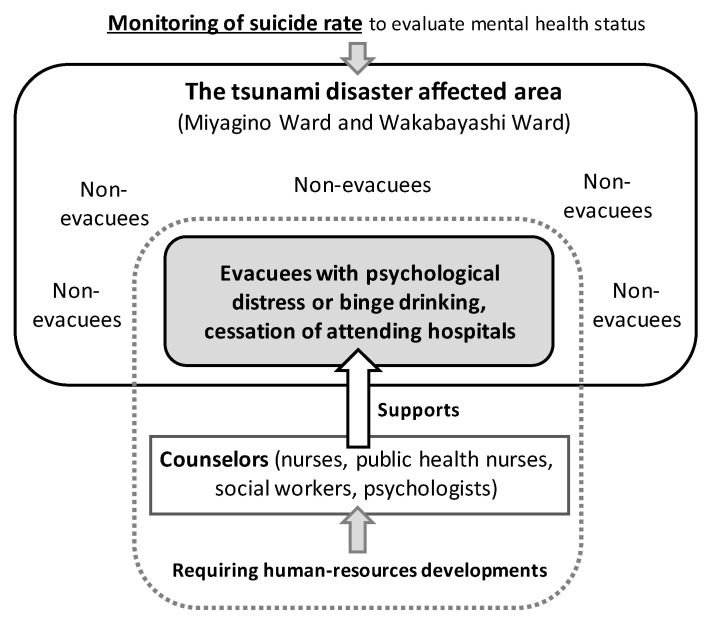
Practical activities of disaster-related mental health interventions.

**Figure 4 ijerph-18-10424-f004:**
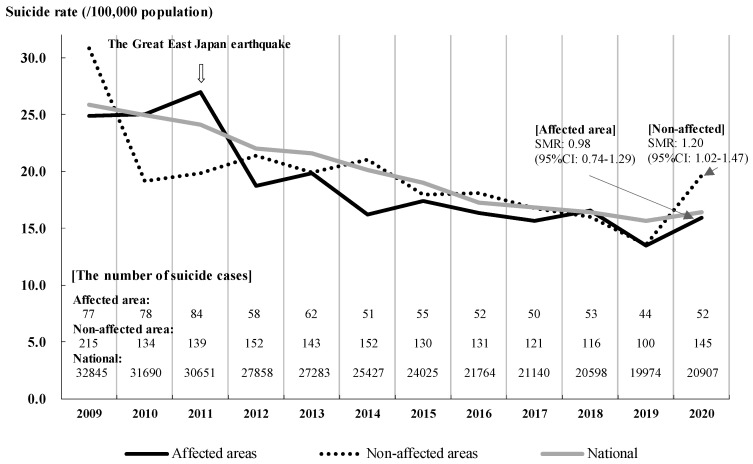
Suicide rate trends in severely affected areas in Sendai City, as compared with the national average. Severely affected area: Miyagino and Wakabayashi Ward in Sendai City were damaged by a tsunami disaster following the GEJE, and there was concentrated support for evacuees through disaster-related mental health interventions. Non-affected areas: Aoba, Taihaku, and Izumi Ward in Sendai City were located in the in-land area. SMR: Standardized Mortality Ratio, comparison between the severely affected area and non-affected area in 2020, and the national average in 2020.

**Table 1 ijerph-18-10424-t001:** Pre-COVID-19 human resource development training seminar series focused on disaster-related mental health interventions.

**A.** Induction seminar regarding disaster-related mental health interventions
**1.** Outline of the GEJE and disaster-related mental interventions **1.1** Evacuation situation of evacuees in shelters and subsequent temporary housing, and the disaster-related mental health intervention history **2.** Traumatic GEJE experiences, and psychological distress and mental health issues among evacuees **2.1** Loss of family, relatives, friends, and housing and employment, and subsequent psychological stress **2.2** Psychological and physical reaction after suffering any stress following a devastating disaster **2.3** Characteristics of unstable psychological reactions over a long-term period among evacuees **3.** Basic tips for disaster-related mental health interventions **3.1** Understanding protective and risk factors for the improvement of evacuees’ mental health issues **3.2** The need for continuous, long-term support—Understanding the fluctuating psychological status among evacuees **3.3** Importance of building a network between other support facilities for the evacuees’ health, life, fi-nancial, and legal issues **3.4** Encouraging the community approach to building a social network and support for the improvement of mental health **4.** Group discussion **4.1** Introducing themselves, and open concerns and anxieties regarding corresponding evacuees’ mental health care
**B.** Case study, group discussion, specific evacuees’ mental health issues, and community approach through disaster-related mental health interventions (five to six times a year *)
**1.** Psychological distress and mental health issues among evacuees **1.1** PTSD, trauma reaction, and grief care **1.2** Alcoholism, evacuees who have alcohol-related problems **1.3** Depressive state, suicide ideation **1.4** Evacuees who have delusions due to dementia or schizophrenia **1.5** Social withdrawal among young and middle-aged evacuees **1.6** Elderly evacuees who have physical and mental health issues **2.** Social and economic problems **2.1** Disadvantaged and disabled evacuees who have difficulty in relocating to public restoration housin **3.** Community approach to build a social network and social support between residents and evacuees **3.1** Community approach to building a social network, and social support and health promotion

***** Case studies and group discussions were held by appropriately selecting topics according to the situation surrounding evacuees, following the disaster.

**Table 2 ijerph-18-10424-t002:** Online training seminar on human resources development regarding disaster-related mental health interventions during the COVID-19 pandemic phase.

**A.** Basic skills for counselors regarding COVID-19 and its relation to mental health issues (twice a year)
**1.** Understanding prevention of COVID-19 infection **1.1** Encouraging basic infection prevention measures (thorough hand washing and wearing of mouth covers, and avoiding closed and close-contact settings) **2.** Coping with evacuees who have any anxiety or concerns about COVID-19 **2.1** Understanding the scenarios caused by the COVID-19 pandemic, and its influence on mental health **2.2** Tips for coping with stress (e.g., maintaining a well-balanced lifestyle, diet, daily interventions, and sleep, maintaining communication with family or friends, regular exercise, avoiding the overuse of media related to COVID-19, and appropriate drinking) **2.3** Understanding the gap in risk perception regarding infection among residents **2.4** Appropriate communication with residents who have severe anxiety and concerns **3.** Encouraging self-care for staff who are addressing COVID-19 prevention **3.1** Psychological and physical reaction after suffering any stress during the COVID-19 pandemic **3.2** Tips for coping with stress (e.g., maintaining a well-balanced lifestyle, diet, daily interventions, and sleep, maintaining communication with colleagues, avoiding the overuse of media related to COVID-19, and appropriate drinking) **4.** Telephone counseling skills as compared to face-to-face counseling **4.1** Advantages of telephone counseling during the COVID-19 pandemic **4.2** Basic skills for telephone counseling (e.g., suppressive tone of voice, having sympathy, gratitude for evacuees’ efforts or pains, back-channeling with appropriate timing, and effective summarizing of talk) **4.3** Assessment of risks of mental health status among evacuees (depressive state, sleep condition, irritability, suicide ideation, alcohol-related problems)
**B.** Online lectures corresponding to specific evacuees’ mental health issues through disaster-related mental health interventions (twice a year **)
**1.** Alcoholism, evacuees who have alcohol-related problems **1.1** Physically damaged and affected by social overuse of alcohol and drugs **1.2** Mechanism of development of addiction **1.3** Recovery from alcoholism and alcohol-related problems **1.4** Importance of collaborating with self-support groups **2.** Social withdrawal among middle-aged evacuees **2.1** Outline of social withdrawal **2.2** Need for appropriate assessment of withdrawal—biological, psychological, and social background **2.3** Support for those who have been withdrawn while taking several steps—individual support, family support, and encouragement of social participation

****** This theme was selected because these were recent mental health topics among evacuees in the GEJE recovery phase, and during the COVID-19 pandemic.

## Data Availability

Publicly available datasets regarding suicide rate trend were analyzed in this study. This data can be found here: [https://www.mhlw.go.jp/stf/seisakunitsuite/bunya/0000140901.html] (in Japanese).

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
