# Peer review of "Practical Report of Disaster-Related Mental Health Interventions Following the Great East Japan Earthquake during the COVID-19 Pandemic: Potential for Suicide Prevention"

_ijerph, 2021, doi:10.3390/ijerph181910424_

Round 1

Reviewer 1 Report

"Practical report of disaster-related mental health activities following the Great East Japan Earthquake under the COVID-19 pandemic: Potential for suicide prevention in the affected area"

Thank you for the opportunity to read this paper on disaster-related mental health in Japan. The topic is unclear of how doe impact of GEJE during the COVID-19 mental health through suicide prevention. The paper is poorly written. Moreover, there are some terms misconceptions, mishandling methods and invalid findings of evidence supports.

Point one:

The paper still need to add more information (e.g., how did you define earthquake, how did you compare the impacts between earthquake and COVID-19 pandemic). This is because the earthquake is a natural disaster, but the COVID-19 is deceased. My question is, how did you conceptualize through practice of difference logical/approach?

Point two:

There are so different terms of mental health uses, for instance, COVID-19 pandemic is so long frustrated, moody, sadness through quality of life, but the earthquake is related to recovery in building capacity.

Point three:

The objectives are unclear, why earthquake under COVID-19 is significant? (unclear why this is the case).

Point four:

* Clarify what/how did you conceptualise through data collection? What kinds of methods to do apply? , How did you collect the data? , Who participated in the study? How did you select? What questions are you asked?

Point five:

Findings, discussion and conclusion are invalid evidence supports. What are new practical implications of the study?

Point six:

The writing in a number of sections has been often, at times clunky, and does not adhere to MDPI grammar guidelines. There need to be significant alterations in sentence structure and grammar in general.

Author Response

Please see the attched file.

Reviewer 2 Report

Thank you for according me the valuable opportunity to review this manuscript. The authors describe mental health interventions associated with the Great East Japan Earthquake during the COVID-19 pandemic. Below are my comments.

GENERAL

The manuscript can be improved with English language editing.

The authors described the interventions as ‘mental health activities’ throughout the manuscript. It will be more appropriate to use ‘intervention’ rather. The entire manuscript should be revised for this diction. For example, in line 13 of the abstract, ‘Disaster-related mental health activities’ can be replaced by ‘Disaster-related mental health interventions’.

ARTICLE TYPE

The manuscript is not a ‘case report’, which often describes a rare presentation of a disease in a SINGLE individual. A practical report is not synonymous with a case report, rather. It can be better described as an ‘Article’. Please review the article types for IJERPH here: https://www.mdpi.com/journal/ijerph/instructions

ABSTRACT

Background (Lines 11-12): This should read: ‘This study aims to describe the ongoing disaster-related mental health interventions…..”

A structured abstract may not be appropriate for the manuscript. For example, the methods and results sections do not actually reflect the methodology of and findings from the study.

TITLE:

The title is wordy and can benefit from revision. It should read: ‘Disaster-related mental health interventions following the Great East Japan Earthquake during the COVID-19 pandemic’. Note the replacement of ‘under’ with ‘during’, and ‘activities’ with ‘interventions’ in the suggested title.

RESULTS

The manuscript is missing a ‘Results Section’. The title in Section 3.0 can be revised to read: ‘Results’. The authors should erase ‘Background of disaster-related mental health activities following the GEJE’.

Author Response

Please see the attched file.

Reviewer 3 Report

Thank you.

Best regards.

Author Response

Please see the attched file.

Round 2

Reviewer 1 Report

Thank you for your revision. All suggestions are revised. A revised version is thoughtful and essentially accessible in its reading.